# Meteorological factors and tick density affect the dynamics of SFTS in jiangsu province, China

**Bin Deng**[1☯], **Jia Rui**[1☯], **Shu-yi Liang**[2☯], **Zhi-feng Li**[2], **Kangguo Li**[1], **Shengnan Lin**[1], **Li Luo**[1], **Jingwen Xu**[1], **Weikang Liu**[1], **Jiefeng Huang**[1], **Hongjie Wei**[1], **Tianlong Yang**[1], **Chan Liu**[1], **Zhuoyang Li**[1], **Peihua Li**[1], **Zeyu Zhao**[1], **Yao Wang**[1], **Meng Yang**[1], **Yuanzhao Zhu**[1], **Xingchun Liu**[1], **Nan Zhang**[2], **Xiao-qing Cheng**[2], **Xiao-chen Wang**[2], **Jian-li Hu**[2]*, **Tianmu Chen**[1]*

1 State Key Laboratory of Molecular Vaccinology and Molecular Diagnostics, School of Public Health, Xiamen University, Xiamen City, People's Republic of China, 2 Department of Acute Infectious Diseases Control and Prevention, Jiangsu Provincial Centre for Disease Control and Prevention, Nanjing, People's Republic of China

☯ These authors contributed equally to this work.
* jshjl@jscdc.cn (JlH); 13698665@qq.com (TC)

**Data Availability Statement:** All relevant data are within the manuscript and its Supporting Information files.

## Abstract

### Background

This study aimed to explore whether the transmission routes of severe fever with thrombocytopenia syndrome (SFTS) will be affected by tick density and meteorological factors, and to explore the factors that affect the transmission of SFTS. We used the transmission dynamics model to calculate the transmission rate coefficients of different transmission routes of SFTS, and used the generalized additive model to uncover how meteorological factors and tick density affect the spread of SFTS.

### Methods

In this study, the time-varying infection rate coefficients of different transmission routes of SFTS in Jiangsu Province from 2017 to 2020 were calculated based on the previous multi-population multi-route dynamic model (MMDM) of SFTS. The changes in transmission routes were summarized by collecting questionnaires from 537 SFTS cases in 2018–2020 in Jiangsu Province. The incidence rate of SFTS and the infection rate coefficients of different transmission routes were dependent variables, and month, meteorological factors and tick density were independent variables to establish a generalized additive model (GAM). The optimal GAM was selected using the generalized cross-validation score (GCV), and the model was validated by the 2016 data of Zhejiang Province and 2020 data of Jiangsu Province. The validated GAMs were used to predict the incidence and infection rate coefficients of SFTS in Jiangsu province in 2021, and also to predict the effect of extreme weather on SFTS.

### Results

The number and proportion of infections by different transmission routes for each year and found that tick-to-human and human-to-human infections decreased yearly, but infections

**Funding:** This study was supported by the Bill & Melinda Gates Foundation (INV-005834) award to TC and "Six One Project" Top Talent Research Plan of Jiangsu High Level Health Talents (JL Hu: LGY2019073) awarded to JLH. The funders had no role in study design, data collection and analysis, decision to publish, or preparation of the manuscript.

**Competing interests:** The authors declare that they have no competing interests.

through animal and environmental transmission were gradually increasing. MMDM fitted well with the three-year SFTS incidence data ($P$<0.05). The best intervention to reduce the incidence of SFTS is to reduce the effective exposure of the population to the surroundings. Based on correlation tests, tick density was positively correlated with air temperature, wind speed, and sunshine duration. The best GAM was a model with tick transmissibility to humans as the dependent variable, without considering lagged effects (GCV = 5.9247E-22, $R^2$ = 96%). Reported incidence increased when sunshine duration was higher than 11 h per day and decreased when temperatures were too high (>28˚C). Sunshine duration and temperature had the greatest effect on transmission from host animals to humans. The effect of extreme weather conditions on SFTS was short-term, but there was no effect on SFTS after high temperature and sunshine hours.

## Conclusions

Different factors affect the infection rate coefficients of different transmission routes. Sunshine duration, relative humidity, temperature and tick density are important factors affecting the occurrence of SFTS. Hurricanes reduce the incidence of SFTS in the short term, but have little effect in the long term. The most effective intervention to reduce the incidence of SFTS is to reduce population exposure to high-risk environments.

## Author summary

Severe fever with thrombocytopenia syndrome (SFTS) is an emerging vector-borne disease caused by SFTS virus. After the first case was detected in China in 2009, SFTS endemic areas have gradually increased, with more than 23 provinces and cities reporting SFTS cases. In order to explore the transmission mechanism of SFTS and explain the impact of meteorological factors and tick density on the transmission routes of SFTS, this study collected SFTS cases data, meteorological data and tick surveillance data in Jiangsu Province from 2017 to 2019 to investigate the study question. The multi-population and multi-route dynamic model established in the previous study was used to calculate the infection rate coefficients of various transmission routes of SFTS in Jiangsu Province, and the generalized additive model was established to further elaborate the influence of SFTS transmission mechanism.

## Introduction

Severe fever with thrombocytopenia syndrome (SFTS) is an infectious disease caused by SFTS virus infection with fever, leukopenia, and thrombocytopenia as the main clinical symptoms. The prevalence of SFTS is mainly in Asia [1], including Japan, South Korea and China. SFTS cases were first detected in the rural areas of Hunan Province and Hubei Province, China in 2009. In early years, the fatality rate of SFTS in China was as high as 30% [2]; however, an in-depth research on SFTS, the average fatality rate in China has gradually decreased to 5.3% from 2010 to 2016. Although the case fatality rate has decreased, its epidemic area has continued to expand. 23 provinces in China reported SFTS cases, and the absolute number of infections increased from 2010 to 2016 [3]. Even in 2017, more than 90% of SFTS cases reported in China and the World Health Organization once listed SFTS as one of the nine major infectious

diseases in the key list, which is one of the most serious public health problems in recent years [4,5].

Previous studies have shown that SFTS is a tick-borne disease, and the environment is one of the important factors influencing SFTS [6]. The high-risk groups of SFTS are mainly the people living in mountainous or hilly rural areas. The most effective way to prevent SFTS is to reduce exposure to ticks [7,8]. At the same time, studies have shown that meteorological factors may affect the spread of SFTS by affecting the growth dynamics of ticks and the interaction between ticks and humans [9]. Many researchers choose statistical models or niche models to find that the occurrence of SFTS is positively related to precipitation, temperature, and altitude [10], and negatively related to atmospheric sub-and wind speed [11,12]. Although previous studies have explored the correlation between meteorological factors and the incidence of SFTS, the research on the causal relationship between meteorological factors and occurrence of diseases and is still little, and there was no research to prove how meteorological factors affect the spread of SFTS. This is known as the most difficult "black box" problem in the spread of infectious diseases. Regarding this problem of SFTS transmission, only a few studies speculate that temperature increase will affect the reproduction rate of ticks and increase the incidence of SFTS [9].

The transmission dynamics model uses the real natural history and reported data of the disease to model, which can simulate the true transmission state of the disease [13–15]. Some researchers have established a seasonal transmission dynamics model of hand, foot, and mouth disease (HFMD), and explored meteorological factors and morbidity and transmissibility the relationship between the two has initially revealed the "black box" problem of the spread of HFMD [16]. In the previous research, we also established the first SFTS multi-population and multi-route dynamic model (MMDM) [17], by fitting the real SFTS report data of Jiangsu Province from 2010 to 2019 through the model, and calculating the infection rate coefficients of the four transmission routes (human-to-human transmission, tick-to-human transmission, environment-to-human transmission and host animal-to-human transmission) over time, which are human-to-human respectively infection rate coefficient ($\beta_1$), tick-to-human infection rate coefficient ($\beta_{21}$), environment-to-human infection rate coefficient ($\beta_{w1}$) and host animal-to-human infection rate coefficient ($\beta_{31}$).

In this study, Jiangsu Province was selected as the study area. And the infection rate coefficients of different transmission routes of SFTS in Jiangsu province in 2017–2020 and in Zhejiang province in 2016 were calculated using MMDM. And the impact of different infection rate coefficients on SFTS incidence was reduced by MMDM simulation. A generalized additive model (GAM) developed SFTS incidence and various infection rate coefficients as dependent variables and meteorological factors and tick density as independent variables. The GAM was used to elaborate how meteorological factors and tick density affect SFTS incidence by influencing which transmission route and thus SFTS incidence. The GAM was used to simulate the effects of different meteorological conditions on SFTS and to uncover the "black box" problem between meteorological factors and SFTS.

## Methods

### Study area

Jiangsu Province is located at the eastern coastal area of mainland China, spanning 30˚45'~35˚ 08' north latitude and 116˚21'~121˚56' east longitude. Jiangsu has a transitional climate from temperate zone to subtropical zone. It has a mild climate, moderate precipitation, and four distinct seasons. It is bounded by the Huaihe River and Subei Irrigation Channel. The north is a warm temperate humid and semi-humid monsoon climate, and the south is a subtropical

humid monsoon climate. The forest area of Jiangsu Province is 1.56 million hectares, and the forest coverage rate is 22.8%.

### Data collection

The 2017–2020 SFTS incidence data in Jiangsu Province came from the Jiangsu Provincial Center for Disease Control and Prevention (CDC). The data is transmittal card data, which includes personal information such as location of onset, date of onset, and gender. We also collected case questionnaires of 537 cases in Jiangsu Province from 2018 to 2020, which included information on the place of residence, history of tick bites, whether or not to keep pets, and whether or not to come into contact with SFTS patients. The 2017–2021 meteorological data of Jiangsu Province came from the China Meteorological Administration, which includes the average wind speed (1m/s), sunshine duration (1h), average temperature (1˚C), and 24-hour precipitation (1mm) in days, average air pressure (1hPa) and relative humidity (1%). The vegetation coverage data of Jiangsu Province came from GlobeLand30.

The data of tick density was monitored by the 2018 Jiangsu Provincial monitoring project, which means the monitor will be carried out once a month from March to October. The cloth flag method was collected for two consecutive mornings, and the monitoring points were mainly around the place where the current cases of vector-borne diseases or the previous cases occurred.

Data on the incidence of SFTS in Zhejiang Province in 2016 and previous meteorological data for Zhejiang Province were obtained from published data.

### Study design and infection rate coefficient calculation

This study was designed as the Fig 1. This study used the established SFTS multi-population and multi-route dynamic model (MMDM) [17] to calculate the infection rate coefficients of SFTS in Jiangsu Province from 2017 to 2020, including the coefficient of human-to-human transmission ($\beta_1$), the coefficient of tick-to-human transmission ($\beta_{21}$), and the coefficient of environmental-to-human transmission rate ($\beta_{w1}$) and the infection rate coefficient of the host animal to humans ($\beta_{31}$). The individual transmission rate coefficients are used to indicate the different transmission routes of SFTS and to reflect the infection dynamics of SFTS by describing the trend change.

Based on the data from the 537 case questionnaire in Jiangsu province in 2018–2020, we defined human-to-human transmission as infected individuals without a history of tick bite but exposed to other patients; tick-to-human transmission as infected individuals with a history of tick bite; environmental-to-human transmission as infected individuals without a history of tick bite and without keeping or contacting animals but living in hilly areas; and animal-to-human transmission as infected individuals without a history of tick bite but who had contact with or kept animals.

Applying the MMDM already established by previous studies [17], the incidence data of Jiangsu Province in 2017–2020 and Zhejiang Province in 2016 were fitted to calculate the transmission rate coefficients of SFTS in different transmission routes in the two provinces by Berkeley Madonna 8.3.18.

### Simulation of interventions

There is no clear and systematic program for prevention and control measures for SFTS. Given that for dengue fever, which is also an insect-borne disease, it mainly includes household survey, health promotion, environmental cleaning, residual spraying and space spraying using insecticides [14]. Meanwhile, the effects of interventions regarding case isolation and

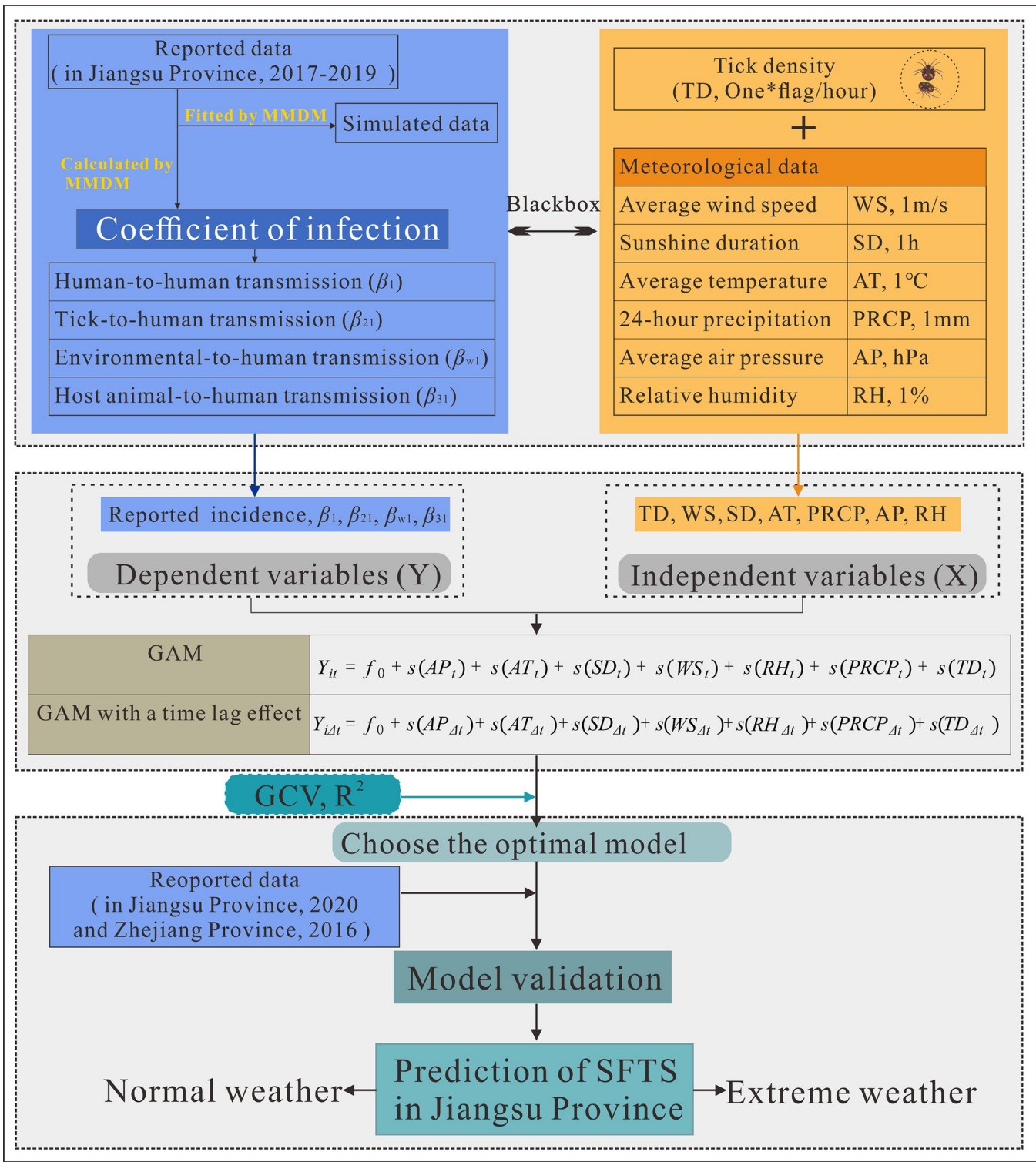

**Fig 1. Study design of meteorological factors and tick density affect the SFTS transmission.** SD = Sunshine duration, RH = Relative humidity, AT = Average temperature, PRCP = 24-hour precipitation, WS = Wind speed, TD = Tick density, and GCV = generalized cross-validation score; GAM: Generalized additive model; MMDM: Multi-population and multi-route dynamic model of SFTS.

treatment with potent drugs have been simulated in previous articles [17]. In this paper, we change the infection rate coefficient by adding the contact efficiency ($\theta$) in MMDM as a way to simulate the effect of controlling the prevention and control of different transmission routes. Different transmission routes have certain contact efficiency, where the contact efficiency between human and human is set as $\theta_1$, between tick and human is set as $\theta_2$, between animal and human is set as $\theta_3$, and between environment and human is set as $\theta_4$.

We assume that the contact efficiency of different transmission routes can be reduced to different levels (10~100%). The impact of SFTS can be controlled by varying the exposure efficiency of a single route of transmission and the impact of multiple routes of transmission by varying the exposure efficiency of multiple routes. According to the literature [14], total attack rate (TAR), absolute effectiveness (AE), and relative effectiveness (RE) will be used as indicators to reflect the effectiveness of the intervention.

## Establishment of the generalized additive models

Previous studies have proved that the relationship between meteorological factors and infectious diseases is complicated. Therefore, this study adopted a GAM, using the original incidence rate (per 1,000,000), $\beta_1$, $\beta_{21}$, $\beta_{w1}$, and $\beta_{31}$ as dependent variables, and using tick density (TD, One*flag/hour), and average wind speed (WS, 1m/s), sunshine hours (SD, 1h), average temperature (AT, 1˚C), 24-hour precipitation (PRCP, 1mm), average air pressure (AP, hPa) and relative humidity (RH, 1%) as independent variables, the complete model is as follows:

$$Y_{it} = f_0 + s(AP_t) + s(AT_t) + s(SD_t) + s(WS_t) + s(RH_t) + s(PRCP_t) + s(TD_t)$$

Among them, $Y_{it}$ represents the value of different dependent variables at day $t$, $i$ can take the reported incidence rate, $\beta_1$, $\beta_{21}$, $\beta_{w1}$, and $\beta_{31}$; $t$ represents the value of the independent variable at time $t$. In addition, most studies believe that there was a time lag effect between the incidence of diseases and meteorological factors, so we constructed a GAM with a time lag effect, as shown below:

$$Y_{i\Delta t} = f_0 + s(AP_{\Delta t}) + s(AT_{\Delta t}) + s(SD_{\Delta t}) + s(WS_{\Delta t}) + s(RH_{\Delta t}) + s(PRCP_{\Delta t}) + s(TD_{\Delta t})$$

Among them, the dependent variable was still the value at time $t$, the value of the independent variable was all 30 days before time $t$, $\Delta t = 30$, and the rest were consistent with the above expression.

## GAM validation and prediction

In this study, we validated the GAM model using meteorological data and tick surveillance data from Jiangsu Province in 2020 and Zhejiang Province in 2016 to prevent overfitting and geographical limitation of the GAM. Subsequently, we used the established and validated GAM to predict the incidence of SFTS in Jiangsu province in 2021, while we predicted the impact of extreme weather on SFTS. Due to the special geographic location of Jiangsu province, the most likely extreme weather includes hurricane and drought, and the two types of weather are set as follows: hurricane = highest historical wind speed (16.8 m/s) * 1.1 + highest historical rainfall (552 mm) * 1.1; drought = highest historical average temperature (35˚C) * 1.1 + longest historical sunshine duration (10) * 1.1. Also based on the maximum rainfall and maximum temperature occurrence point, the hurricane period was set to start on April 30, 2020 and last for 7 days; the drought was set to start on August 30, 2020 and last for 30 days. We used GAM to predict changes in SFTS incidence at the time of extreme weather, one month after the occurrence, and one year after the occurrence as a way to assess short- and

long-term effects, with key outcome indicators including TAR and cumulative number of cases.

### Data analyze

We used Berkeley Madonna 8.3.18 to calculate the transmission rate coefficients and intervention simulations. The correlation tests between independent variables were performed using Spearman's test, and correlation coefficients greater than 0.7 were chosen with caution. We used the "MGCV" library in R 3.2.3 to build the GAM and make predictions; the GAM was estimated using the great likelihood method; different GAMs were selected based on the lowest generalized cross-validation (GCV) score. Image production was done using R 3.2.3 "ggplot2" and table production was done using EXCEL 2019.

## Results

### Characteristics of epidemiological and meteorological factors

In the part of epidemiology of SFTS in Jiangsu Province, the analysis of the months and years of SFTS onset revealed a strict seasonal variation in SFTS onset. We combined the incidence of SFTS with the land use map of Jiangsu Province, as shown in Fig 2. We found that most of the cases were concentrated in the southwestern region of Jiangsu Province, including Nanjing City and Huai'an City, and in areas with high vegetation coverage, such as areas with more broad-leaved forests and shrubs, while water bodies and residential areas have relatively fewer cases. Additionally, we found that the incidence of SFTS had an upward trend in the past three years, with the highest months of incidence mostly occurring in summer. According to the epidemic curve, the peak incidence of SFTS in Jiangsu Province was mainly in summer and autumn, and the incidence rate was higher in autumn than in summer (S1A Fig).

Based on the results of the SFTS case questionnaire in Jiangsu Province for 2018–2019, we collated the number and proportion of infections by different transmission routes for each year and found that tick-to-human and human-to-human infections decreased as the years increased, but infections through animal and environmental transmission were gradually increasing. See Table 1 for details.

In the part of meteorological factors in Jiangsu Province, during the study period, the highest temperature in Jiangsu Province was 34°C, the lowest temperature was -3.5°C, and the average temperature was 16.3°C (S2D Fig); the average air pressure was 987~1040hpa, the sunshine duration was 0~12h (S2C Fig), the relative humidity was 31~98% (S2A Fig), and the precipitation was 0~67mm (S2B Fig), wind speed was 0.7~6.3m/s (S2F Fig), tick density was 0~68*flag/hour (S2E Fig).

### Transmission dynamics model fitting and interventions simulation

As the Fig 3 shown, the MMDM fitted well with SFTS incidence data from 2017–2019 in Jiangsu Province ($P<0.05$). The model fitting results found that the coefficient of infection of environment to human ($\beta_{w1}$) was increasing year by year (S1D Fig), and it was found that the coefficient of infection of human to human ($\beta_1$) was very low in 2018 (S1B Fig).

We reduced the effective exposure rates for the different transmission routes to varying degrees to simulate interventions such as wearing protective clothing, insecticides and disinfection. As shown in Table 2, we found that only the reduction of the environmental-to-human infection rate factor had the greatest impact on the incidence of SFTS, and when the effective contact rate was reduced by 50%, the number of SFTS patients was correspondingly reduced by half.

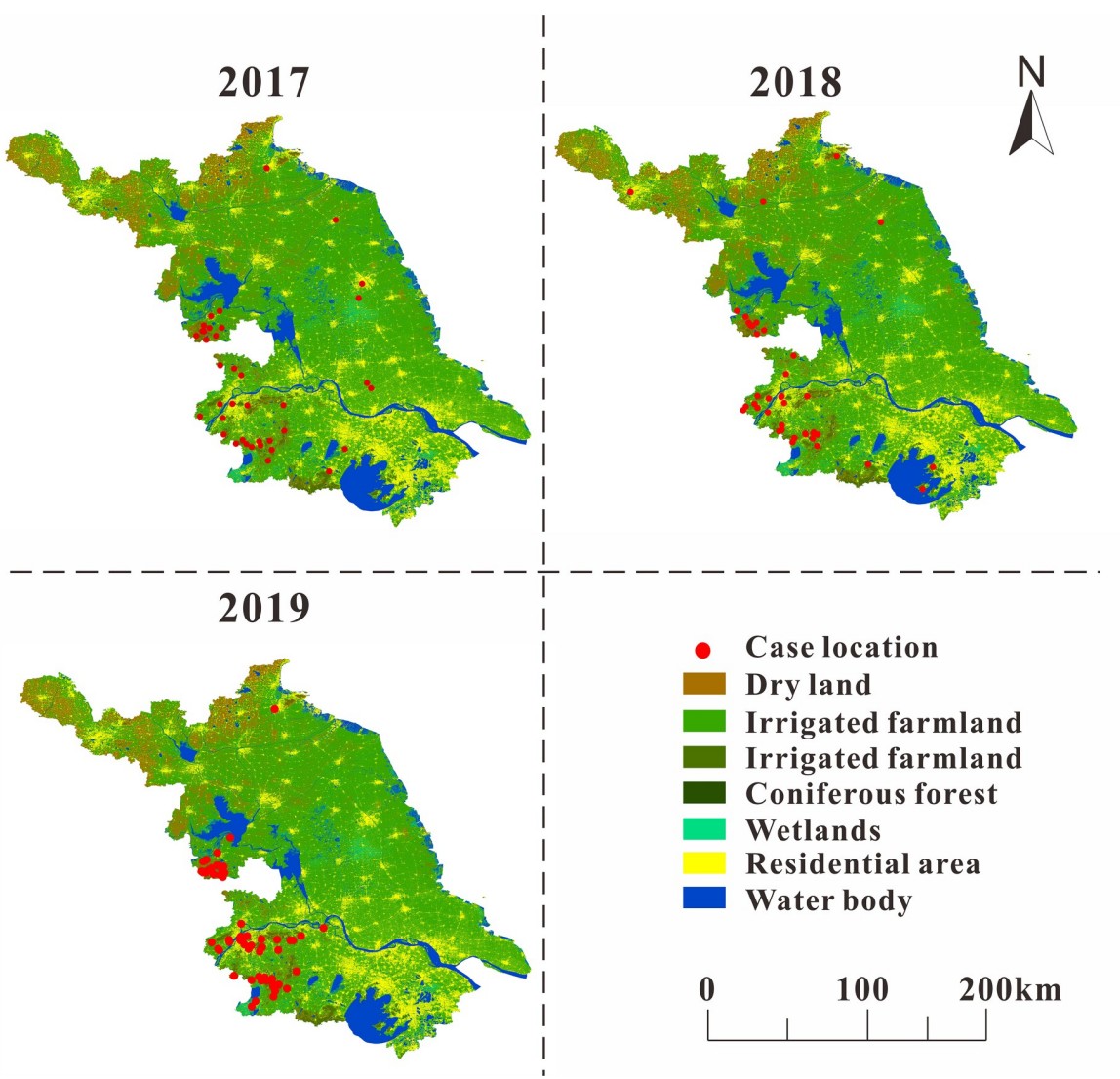

**Fig 2. Land use map of each SFTS case location in Jiangsu Province from 2017 to 2019.** http://www.globallandcover.com/defaults.
html?type=data&src=/Scripts/map/defaults/browse.html&head=browse&type=data (Map source).

## Correlation test between meteorological factors and tick density

According to the Spearman correlation test results of meteorological factors and tick density, we found that air pressure was negatively correlated with other factors, and the correlation with temperature was relatively high ($r = 0.899$). At the same time, it was found that the duration

**Table 1. Transmission routes of infection in 537 SFTS patients in Jiangsu Province, 2018–2020.**

| Year | Tick to human transmission | Human to human transmission | Animal to human transmission | Environment to human transmission | Unclassified | Total (%) |
|------|----------------------------|-----------------------------|------------------------------|-----------------------------------|--------------|-----------|
| 2018 | 37(45.67%) | 6(7.40%) | 20(24.69%) | 10(12.34%) | 8(9.9%) | 81(100%) |
| 2019 | 62(46.97%) | 3(2.27%) | 35(26.52%) | 22(16.67%) | 10(7.58%) | 132 (100%) |
| 2020 | 109(33.64%) | 13(4.01%) | 95(29.32%) | 82(25.31%) | 25(7.72%) | 324 (100%) |

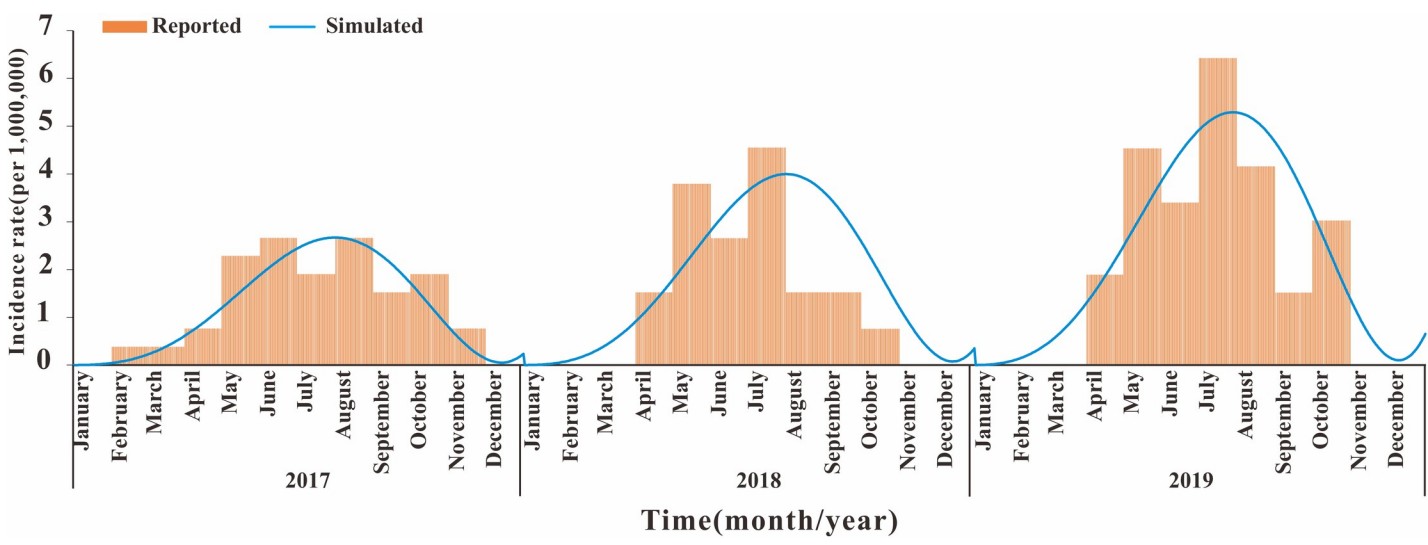

**Fig 3. MMDM fitting result of SFTS incidence of 2017–2019.** MMDM: Multi-population and multi-route dynamic model of SFTS.

of sunshine was positively correlated with temperature and tick density, and negatively correlated with other factors, and the correlation was small. Humidity was positively correlated with temperature, rainfall, and month, and has no correlation with tick density. There was no correlation between air temperature and wind speed, but a positive correlation with rainfall, tick density, etc. Although precipitation has no correlation with tick density, it was positively correlated with wind speed and month. The density of ticks was positively correlated with air temperature, wind speed and sunshine duration, but negatively correlated with air pressure, and the correlation was high ($r$ = -0.51), as shown in Fig 4. After lagging the meteorological factors for 30 days, the results of Spearman test were shown in Fig 5, and there was no significant difference in the results. The two sets of Spearman correlation test results showed that the correlation between average air temperature and average air pressure was too high ($r$ = 0.897, $P$<0.01). After adding the average air pressure and average air temperature into the GAM separately, it was found that the average air pressure was meaningless, so we eliminated it from the model.

## Construction of generalized additive model

To establish a GAM, the average temperature, relative humidity, sunshine duration, precipitation, tick density, month and different dependent variables were used. According to the GCV score and $R^2$, the models with the lowest GCV scores for the different dependent variables among all the GAMs we developed are shown in Table 3. According to the results in Table 3, we can find that for GAMs, the GCV scores of the models with 30-day lags are generally higher than those of the models without lags, and also the model fit is better for the models without lags. For tick density and meteorological factors, the most suitable dependent variable was the coefficient of infection rate of tick-to-human, which had the lowest GCV scores (GCV = 5.9247E-22, $R^2$ = 96%).

## The relationship among SFTS reported incidence, meteorological factors, and tick density

For the reported incidence, in the no time lag GAM, all factors were nonlinear except for relative humidity, precipitation and wind speed. Among them, the effect of sunshine duration on incidence was not significant, but there was an increasing trend when the sunshine duration

**Table 2.  Simulation of interventions for SFTS by the multi-population and multi-route dynamics model.**

| Intervention | Number of cases | TAR | AE | RE (%) |
|---|---|---|---|---|
| H-H-T-R* | | | | |
| 10% | 1140 | 4.32E-05 | 0.00E+00 | 0.00% |
| 20% | 1140 | 4.32E-05 | 2.28E-14 | 0.00% |
| 30% | 1140 | 4.32E-05 | 4.55E-14 | 0.00% |
| 40% | 1140 | 4.32E-05 | 6.83E-14 | 0.00% |
| 50% | 1140 | 4.32E-05 | 9.11E-14 | 0.00% |
| 60% | 1140 | 4.32E-05 | 1.14E-13 | 0.00% |
| 70% | 1140 | 4.32E-05 | 1.37E-13 | 0.00% |
| 80% | 1140 | 4.32E-05 | 1.59E-13 | 0.00% |
| 90% | 1140 | 4.32E-05 | 1.82E-13 | 0.00% |
| 100% | 1140 | 4.32E-05 | 2.05E-13 | 0.00% |
| T-H-T-R* | | | | |
| 10% | 1140 | 4.32E-05 | 0.00E+00 | 0.00% |
| 20% | 1140 | 4.32E-05 | 7.85E-16 | 0.00% |
| 30% | 1140 | 4.32E-05 | 1.57E-15 | 0.00% |
| 40% | 1140 | 4.32E-05 | 2.35E-15 | 0.00% |
| 50% | 1140 | 4.32E-05 | 3.14E-15 | 0.00% |
| 60% | 1140 | 4.32E-05 | 3.92E-15 | 0.00% |
| 70% | 1140 | 4.32E-05 | 4.71E-15 | 0.00% |
| 80% | 1140 | 4.32E-05 | 5.49E-15 | 0.00% |
| 90% | 1140 | 4.32E-05 | 6.28E-15 | 0.00% |
| 100% | 1140 | 4.32E-05 | 7.06E-15 | 0.00% |
| A-H-T-R* | | | | |
| 10% | 1140 | 4.32E-05 | 0.00E+00 | 0.00% |
| 20% | 1140 | 4.32E-05 | 1.05E-09 | 0.00% |
| 30% | 1140 | 4.32E-05 | 2.09E-09 | 0.00% |
| 40% | 1140 | 4.32E-05 | 3.14E-09 | 0.01% |
| 50% | 1139 | 4.32E-05 | 4.19E-09 | 0.01% |
| 60% | 1139 | 4.32E-05 | 5.24E-09 | 0.01% |
| 70% | 1139 | 4.32E-05 | 6.28E-09 | 0.01% |
| 80% | 1139 | 4.32E-05 | 7.33E-09 | 0.02% |
| 90% | 1139 | 4.32E-05 | 8.38E-09 | 0.02% |
| 100% | 1139 | 4.32E-05 | 9.43E-09 | 0.02% |
| E-H-T-R* | | | | |
| 10% | 1140 | 4.32E-05 | 0.00E+00 | 0.00% |
| 20% | 1026 | 3.89E-05 | 4.32E-06 | 10.00% |
| 30% | 912 | 3.45E-05 | 8.63E-06 | 20.00% |
| 40% | 798 | 3.02E-05 | 1.30E-05 | 29.99% |
| 50% | 684 | 2.59E-05 | 1.73E-05 | 39.99% |
| 60% | 570 | 2.16E-05 | 2.16E-05 | 49.99% |
| 70% | 456 | 1.73E-05 | 2.59E-05 | 59.99% |
| 80% | 342 | 1.30E-05 | 3.02E-05 | 69.98% |
| 90% | 228 | 8.64E-06 | 3.45E-05 | 79.98% |
| 100% | 114 | 4.33E-06 | 3.88E-05 | 89.98% |
| H-H-T-R100%+A-H-T-R100%+T-H-T-R100% | 1139 | 4.32E-05 | 9.43E-09 | 0.02% |

*TAR: Total attack rate; AE: Absolute effectiveness; RE: Relatively effectiveness; H-H-T-R: Human to human transmission rate; E-H-T-R: Environment to human transmission rate; T-H-T-R: Tick to human transmission rate; A-H-T-R: Animal to human transmission rate

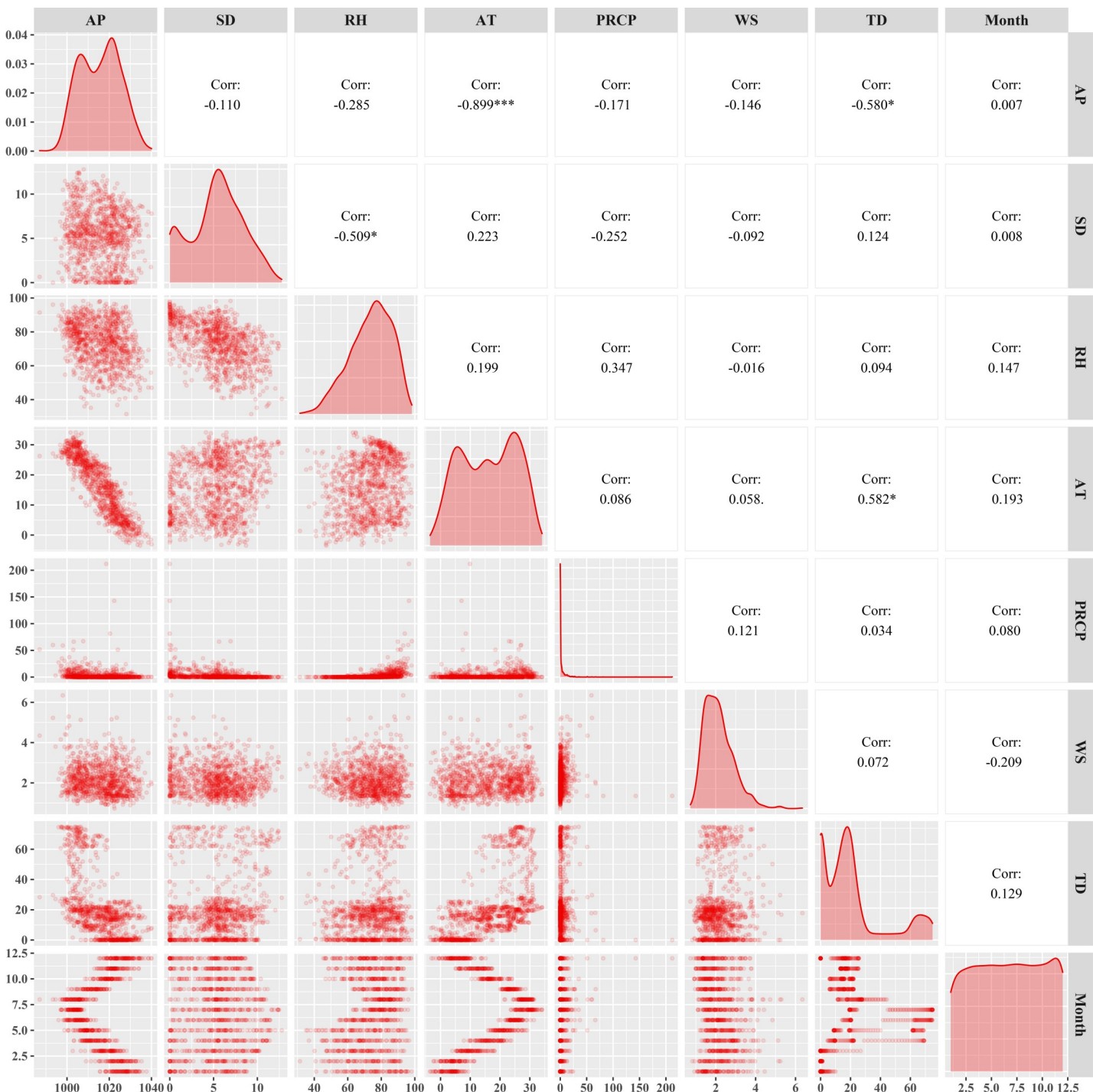

**Fig 4. Correlation analysis between no time lagging meteorological factors and tick density.** AP = Air pressure; SD = Sunshine duration; RH = Relative humidity; AT = Average temperature; PRCP = 24-hour precipitation; WS = Wind speed; TD = Tick density. Correlation coefficient (r) greater than 0.7 indicates a strong correlation between the two.

was too high (>11 h); the incidence decreased rapidly when the average temperature was higher than about 26°C; the effect of tick density on incidence was positive, and the incidence showed an increasing trend as the tick density increased; for months, the incidence of SFTS was highest

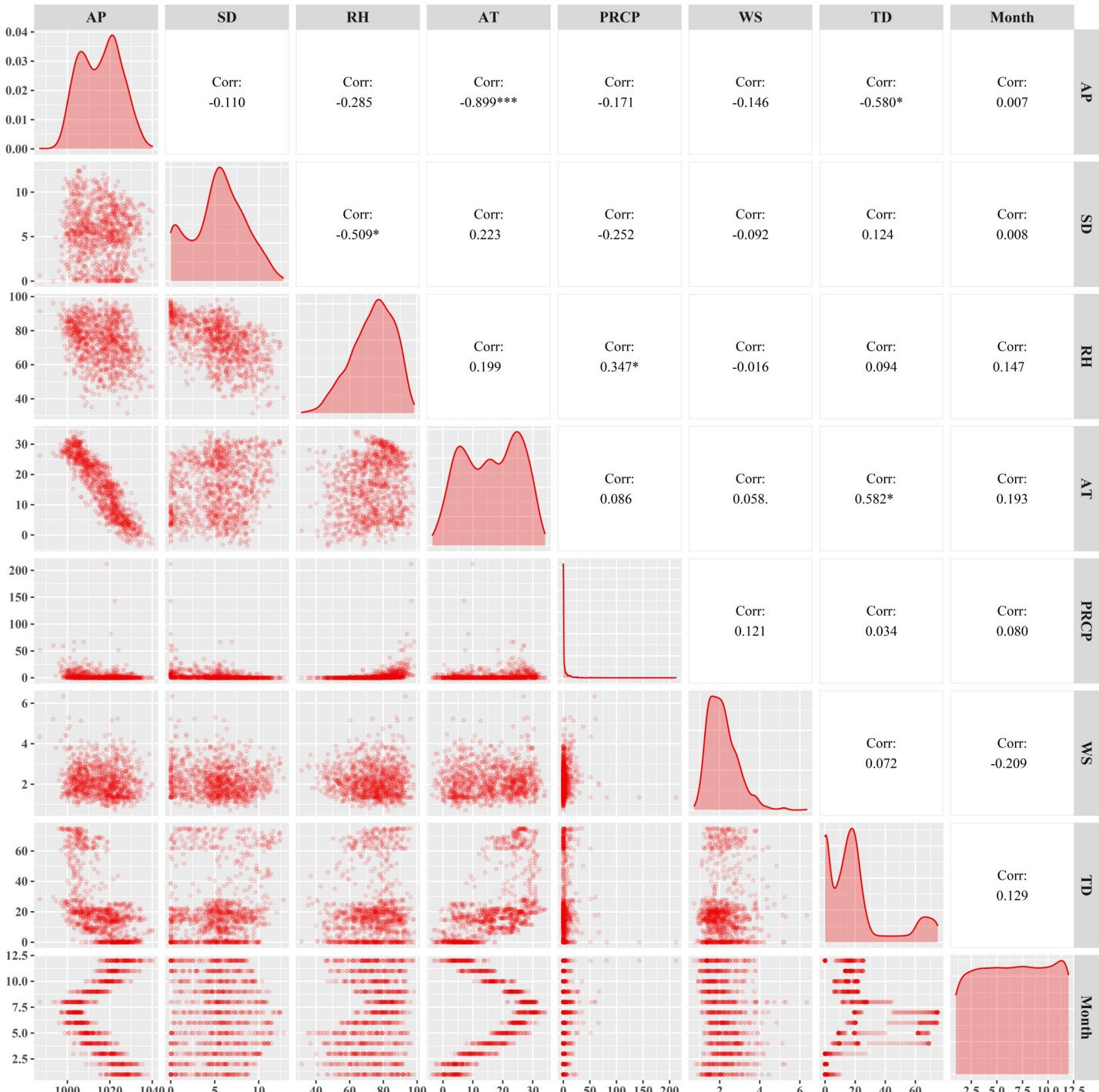

**Fig 5. Correlation analysis between time lagging meteorological factors and tick density.** AP = Air pressure; SD = Sunshine duration; RH = Relative humidity; AT = Average temperature; PRCP = 24-hour precipitation; WS = Wind speed; TD = Tick density. Correlation coefficient (r) greater than 0.7 indicates a strong correlation between the two.

in May to September (Figs 6A1 and S3A). In the 30-day lagged GAM, only mean temperature and month were significant (P<0.05), where the effect of month was the same as non-lagged, but the effect of mean temperature on incidence was attenuated (Figs 6B1 and S3B).

**Table 3. Optimal generalized additive models built with time lags and non-time lags.**

| Model type | Formula | GCV | $R^2$ |
|---|---|---|---|
| **No time lag model** | | | |
| Dependent variables | | | |
| Reported incidence | $f0+s(SD_t) + RH_t + s(AT_t) + PRCP_t + s(Month_t) + WS_t + s(TD_t)$ | 4.05E-15 | 86% |
| Coefficient of infection of human to human ($\beta_1$) | $f0+SD_t + RH_t + s(AT_t) + PRCP_t + s(Month_t) + s(WS_t) + s(TD_t)$ | 1.59E-19 | 62% |
| Coefficient of infection of environment to human ($\beta_{w1}$) | $f0+SD_t + s(RH_t) + s(AT_t) + s(PRCP_t) + s(Month_t) + s(WS_t) + s(TD_t)$ | 9.04E-18 | 86% |
| Coefficient of infection of tick ($\beta_{21}$) | $f0+SD_t + RH_t + s(AT_t) + PRCP_t + s(Month_t) + s(WS_t) + s(TD_t)$ | 5.92E-22 | 96% |
| Coefficient of infection of animal to human ($\beta_{31}$) | $f0+SD_t + s(RH_t) + s(AT_t) + s(PRCP_t) + s(Month_t) + s(WS_t) + s(TD_t)$ | 3.33E-18 | 75% |
| Time lag 30 days model | | | |
| Dependent variables | | | |
| Reported incidence | $f0+SD_{\Delta t} + RH_{\Delta t} + s(AT_{\Delta t}) + PRCP_{\Delta t} + s(Month_{\Delta t}) + WS_{\Delta t} + TD_{\Delta t}$ | 4.87E-15 | 83% |
| Coefficient of infection of human to human ($\beta_1$) | $f0+s(SD_{\Delta t}) + s(RH_{\Delta t}) + s(AT_{\Delta t}) + s(PRCP_{\Delta t}) + s(Month_{\Delta t}) + s(WS_{\Delta t}) + TD_{\Delta t}$ | 1.76E-19 | 58% |
| Coefficient of infection of environment to human ($\beta_{w1}$) | $f0+s(SD_{\Delta t}) + s(RH_{\Delta t}) + s(AT_{\Delta t}) + PRCP_{\Delta t} + s(Month_{\Delta t}) + s(WS_{\Delta t}) + s(TD_{\Delta t})$ | 8.36E-18 | 87% |
| Coefficient of infection of tick ($\beta_{21}$) | $f0+s(SD_{\Delta t}) + s(RH_{\Delta t}) + s(AT_{\Delta t}) + PRCP_{\Delta t} + s(Month_{\Delta t}) + s(WS_{\Delta t}) + TD_{\Delta t}$ | 6.05E-22 | 96% |
| Coefficient of infection of animal to human ($\beta_{31}$) | $f0+s(SD_{\Delta t}) + s(RH_{\Delta t}) + s(AT_{\Delta t}) + PRCP_{\Delta t} + s(Month_{\Delta t}) + s(WS_{\Delta t}) + TD_{\Delta t}$ | 3.69E-18 | 72% |

*SD = Sunshine duration, RH = Relative humidity, AT = Average temperature, PRCP = 24-hour precipitation, WS = Wind speed, TD = Tick density, and GCV = generalized cross-validation score

## The relationship among human-to-human transmission, meteorological factors, and tick density

As shown in Fig 6A2, for the coefficient of infection rate of human-to-human, the effects of average temperature, month, wind speed and tick density were all nonlinear in the GAM with no time lag. However, wind speed had a smaller effect on inter-population; while temperature was more pronounced for transmission among populations, which was reduced when temperatures were too low (<5˚C) or too high (>25˚C). The effect of tick density on incidence was negative, with population-to-population transmission decreasing as tick density increased; month had only a peak for between-population (S4A Fig). In the 30-day lagged GAM, all meteorological factors except tick density had a nonlinear effect on transmission among populations, with the largest changes compared to the model with no time lag being that relative humidity and sunshine duration had a negative effect on transmission among populations, and that transmission of SFTS among populations decreased as both increased, with no differences in the remaining factors (Figs 6B2 and S4B).

## The relationship among environment-to-human transmission, meteorological factors, and tick density

For the coefficient of infection of the environment-to-human, in the GAM without time lag, the effects of all factors were non-linear except for the sunshine duration, and the effect of average temperature was not significant ($P > 0.05$). Among the remaining factors, the most

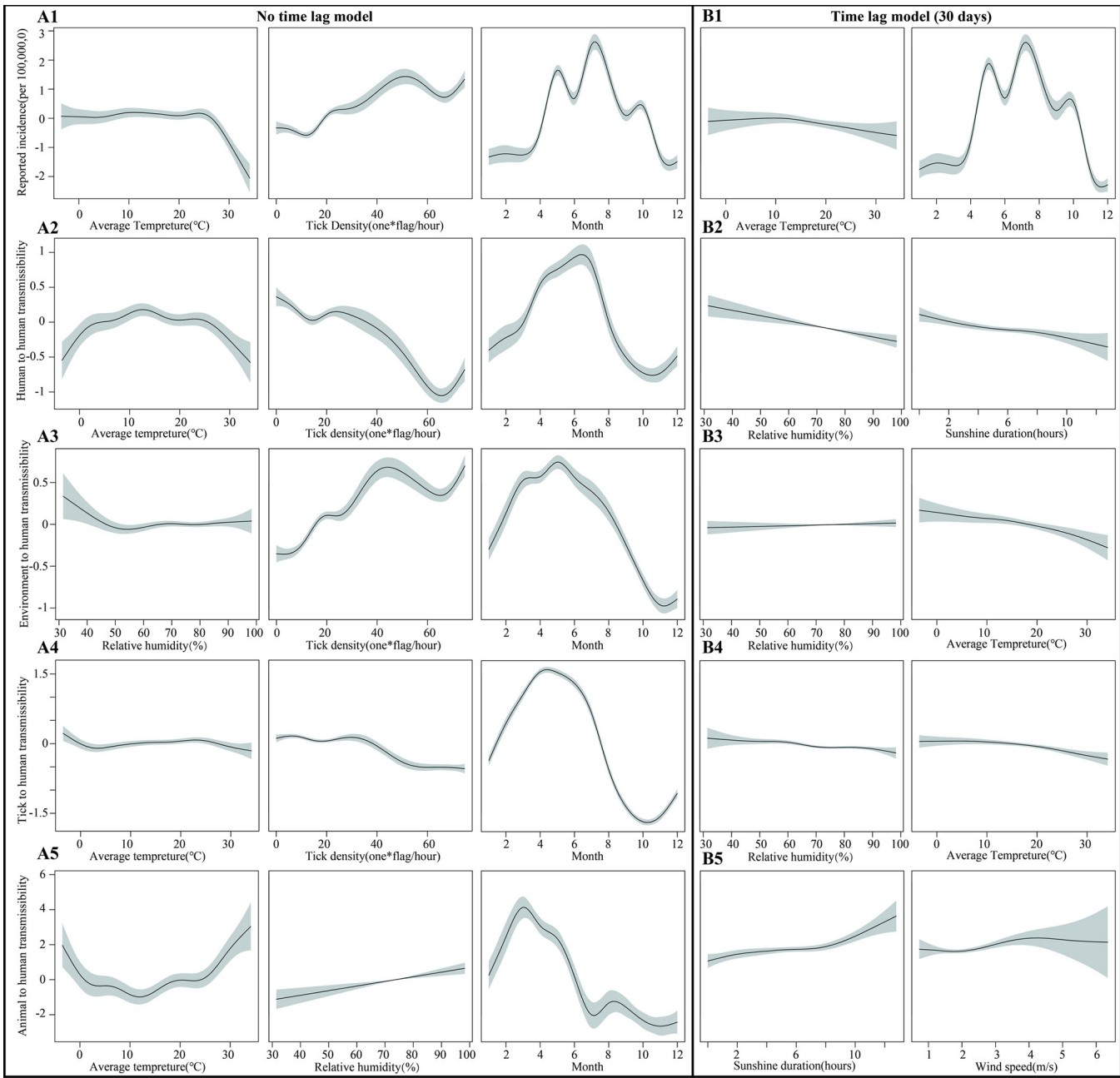

**Fig 6. Non-linear relationship between SFTS incidence and different transmissibility with meteorological factors and tick density in Jiangsu Province.**
Part A: SFTS incidence and different infection coefficients with meteorological factors and tick density in no time lag GAM; A1: Plots of non-linear relationship with factors associated with reported incidence; A2: Plot of non-linear relationship with factors associated with the infection coefficient of human-to-human; A3: Plot of non-linear relationship with factors associated with the infection coefficient of environment-to-human; A4: Plot of non-linear relationship with factors associated with the infection coefficient of tick-to-human; A5: Plot of non-linear relationship with factors associated with the infection coefficient of animal-to-human. Part B: SFTS incidence and different infection coefficients with meteorological factors and tick density in time lag GAM; B1: Plots of non-linear relationship with factors associated with reported incidence; B2: Plot of non-linear relationship with factors associated with the infection coefficient of human-to-human; B3: Plot of non-linear relationship with factors associated with the infection coefficient of environment-to-human; B4: Plot of non-linear relationship with factors associated with the infection coefficient of tick-to-human; B5: Plot of non-linear relationship with factors associated with the infection coefficient of animal-to-human).

important factor is the effect of tick density, which gradually increases the transmission of the environment as tick density increases; for relative humidity, the transmission of the environment decreases as relative humidity increases, with less effect when relative humidity is greater than 50%. The effect of month for environmental transmission is consistent with human-to-human transmission, with only one peak (Figs 6A3 and S5A). Compared to the model with no time lag, the larger differences in the GAM with a 30-day lag are the effects from relative humidity and average temperature. The time lag attenuates the effect of relative humidity, but the transmission in the environment gradually decreases as the average temperature increases (Figs 6B3 and S5B).

## The relationship among tick-to-human transmission, meteorological factors, and tick density

For the tick-to-human transmission coefficient, only the effects of average temperature, tick density and month were nonlinear in the GAM without a time lag, but the effects of all factors were small, with tick-to-human transmission decreasing gradually, but to a lesser extent, as temperature and tick density increased (Figs 6A4 and S6A). After lagging the time by 30 days, relative humidity also affects tick-to-human transmission, which gradually decreases with increasing relative humidity, but to a lesser extent, and the remaining factors do not differ significantly, as detailed in Figs 6B4 and S6B.

## The relationship among animal-to-human transmission, meteorological factors, and tick density

For the animal-to-human infection coefficient, in the GAM without time lag, all factors except relative humidity affect transmission. Among them, the effect of mean temperature is opposite to the transmission between populations, with a trend of decreasing and then increasing animal-to-human transmission as the temperature increases, and the weakest animal-to-human transmission occurs when the temperature is between 10 and 16˚C. Also, the month makes a difference for animal-to-human transmission, which decreases rapidly after April (Figs 6A5 and S7A). Among the GAMs with a time lag of 30 days, the greatest effects were sunshine duration and wind speed, with animal-to-human transmission gradually increasing as both increased (Figs 6B5 and S7B).

## No time lag GAM validation and prediction

Based on GCV scores, $R^2$ and the effect of different factors, we chose no time lag GAM to predict the change in incidence in future and extreme weather conditions. We found that the established GAM predicted better results for the reported incidence of SFTS and individual infection rate coefficients in Zhejiang Province in 2016, and the predicted data differed less from the real data (Fig 7A–7E).

In Fig 8 and Table 4, we predicted the incidence rate and the transmission rate coefficients of each transmission route in Jiangsu Province from 2020 to 2021 using the GAM we established, and also verified again using the incidence rate reported in Jiangsu Province and the transmission rate coefficients calculated by MMDM, and found that the GAM predictions were better except for tick-to-human transmission. In Fig 8, we predicted the effect of SFTS on incidence and transmission in hurricane occurrence, respectively. As shown in Table 4, for incidence, SFTS incidence is substantially reduced during the hurricane, bringing the number of incidences down from 229 to 0; however, the hurricane has less impact on SFTS incidence one month and one year later afterwards. For different transmission rate coefficients, all

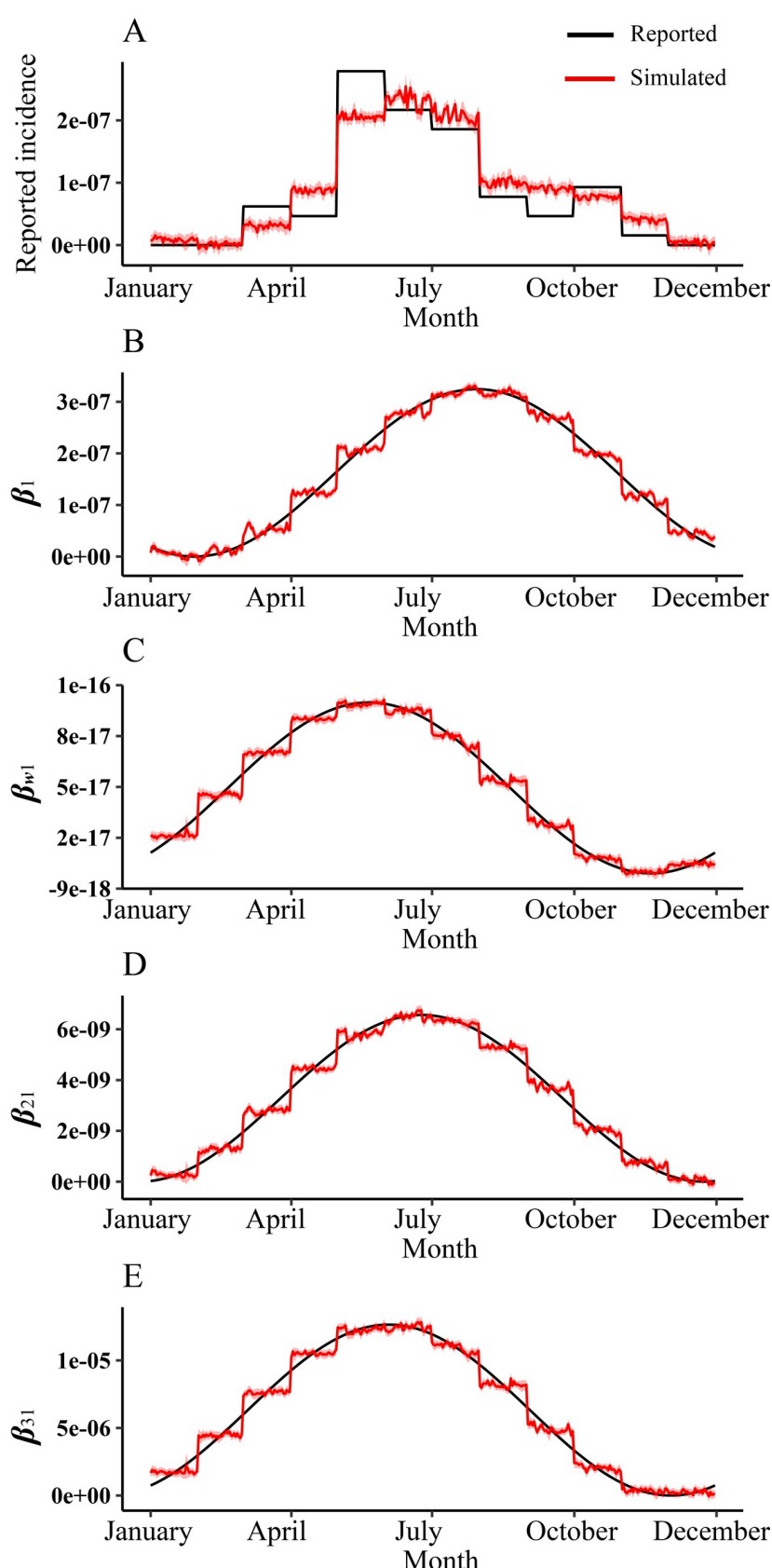

**Fig 7. Comparison of reported data of SFTS incidence and different infection coefficients with predicted GAM data without time lag in Zhejiang Province, 2016.** A: Reported incidence and GAM simulated incidence; B: Calculated human-to-human infection coefficient and the GAM simulated infection coefficient, $\beta_1$:Infection coefficient of human-to-human; $\beta_{w1}$:Infection coefficient of environment-to-human; $\beta_{21}$:Infection coefficient of tick-to-human; $\beta_{31}$:Infection coefficient of animal-to-human).

transmission rate coefficients were reduced at the time of the hurricane. And for transmission in 2021, each transmission rate factor is reduced compared to when no hurricane occurs. In Fig 8, we predict the impact of SFTS on incidence and transmission in the event of drought, respectively. As shown in Table 4 and Fig 8, for incidence and each infection rate, there was little effect of drought either at that time period.

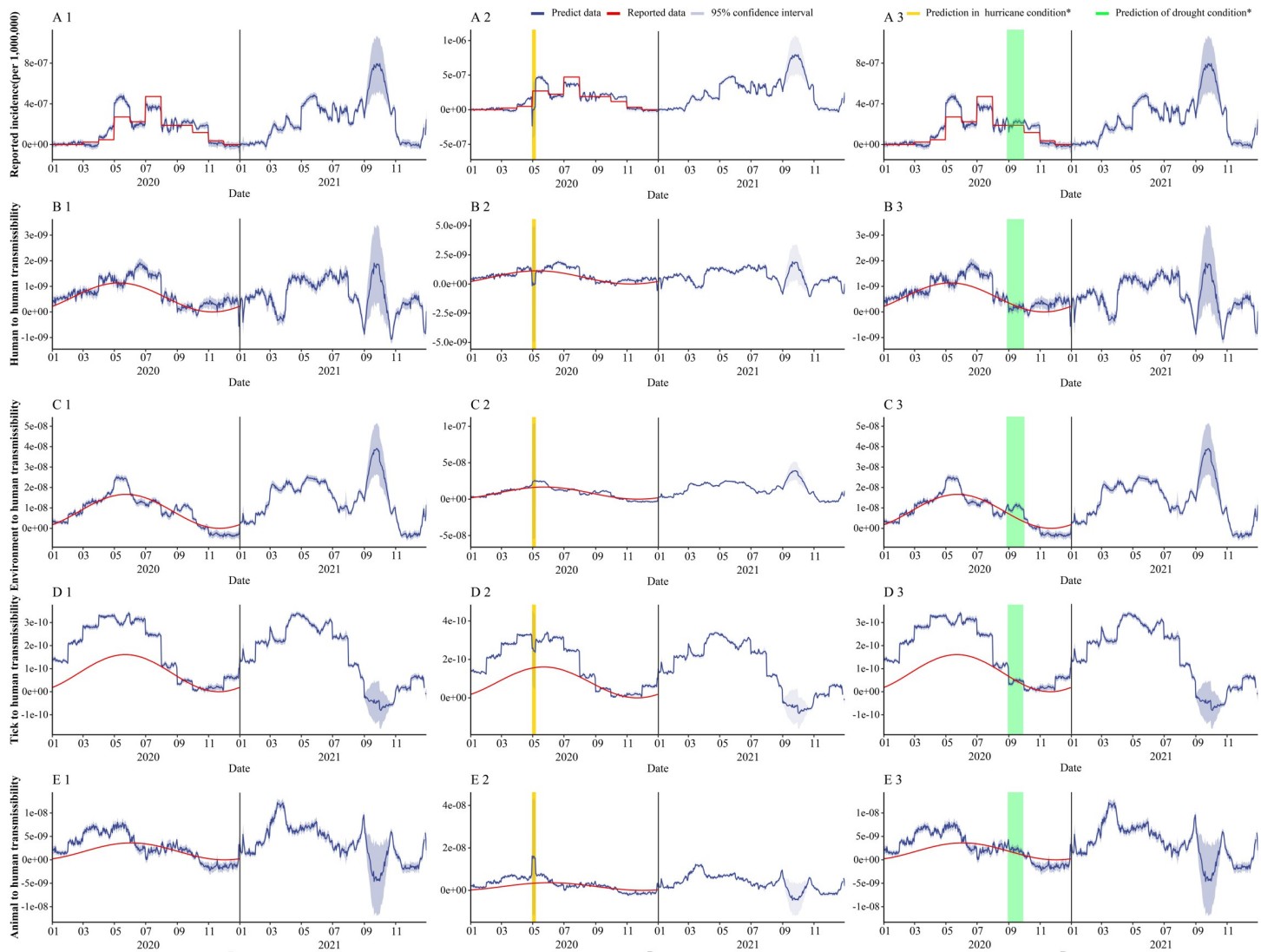

**Fig 8. GAM predicts the incidence of SFTS, infection coefficient and the impact of extreme weather in Jiangsu Province in 2020–2021.** Part 1: the prediction of GAM about SFTS incidence and different infection coefficients in normal weather; Part 2: the prediction of GAM about SFTS incidence and different infection coefficients in hurricane; Yellow part is the duration of the hurricane; Part 3: the prediction of GAM about SFTS incidence and different infection coefficients in drought; green part is the duration of the drought).

Table 4. **Extreme weather forecast in Jiangsu Province.**

| Condition | Stage* | TAR | Cases |
|---|---|---|---|
| Normal weather | stage 1 | 2.71E-06 | 229 |
| | stage 2 | 1.23E-05 | 1044 |
| | stage 3 | 6.03E-05 | 5111 |
| Hurricane* | stage 1 | 0.00E+00 | 0 |
| | stage 2 | 1.23E-05 | 1044 |
| | stage 3 | 6.03E-05 | 5111 |
| Normal weather | stage 4 | 6.65E-06 | 564 |
| | stage 5 | 5.76E-06 | 488 |
| | stage 6 | 8.05E-05 | 6819 |
| Drought* | stage 4 | 6.65E-06 | 564 |
| | stage 5 | 5.76E-06 | 488 |
| | stage 6 | 8.05E-05 | 6819 |

*Hurricane: highest historical wind speed (16.8 m/s) * 1.1 + highest historical precipitation (552 mm) * 1.1;
Drought = highest historical average temperature (35˚C) * 1.1 + longest historical sunshine duration (10) * 1.1; stage 1: 2020.04.30–2020.05.06; stage 2: 2020.05.07–2020.06.07; stage 3: 2020.04.30–2021.04.30; stage 4: 2020.08.30–2020.09.30; stage 5: 2020.10.01–2020.10.30; stage 6: 2020.10.01–2021.10.01

## Discussion

Although many studies have explored the relationship between meteorological factors and SFTS incidence, they are limited to correlation analysis and cannot reveal how meteorological factors affect SFTS. this study first used the established multiple population multiple route transmission dynamics model (MMDM) to calculate the infection rate coefficients of SFTS by different transmission routes. The effect of meteorological factors and tick surveillance density on SFTS transmission was investigated by combining meteorological factors and tick density with reported data and infection rate coefficients of SFTS through generalized additive model (GAM).This study found that the incidence of SFTS had gradually increased in recent years, and most of the cases occurred in places with high vegetation coverage, which was consistent with the characteristics of places with high incidence of SFTS revealed in previous studies [11,18]. However, it should be noted that most of the SFTS incidence data collected in this study were concentrated in Nanjing City and Huai'an City, Jiangsu Province. SFTS cases in other areas were relatively rare. This may be due to the low population density in the local area or the incomplete SFTS monitoring and reporting system [12,19].

In this study, we found a sharp increase in the incidence of SFTS when the sunshine hours exceeded 10 h. Previous studies have also shown that sunshine hours are a key factor in the occurrence of SFTS [6], but it is believed that sunshine hours are an intermediate function of temperature that affects SFTS and does not independently affect SFTS. The results of this study showed that the length of daylight hours severely affects the infection rate coefficient of animal transmission, with a rapid increase in animal-to-human infection rate when daylight hours are greater than 8 hours. The reason for this may be that with increasing daylight hours, the prolonged stay of host animals (e.g., goats) outdoors leads to an increased chance of animal or human exposure to ticks and animal excreta, leading to an increased incidence of SFTS [20,21]. In addition, the length of daylight also affects human-to-human transmissibility. $\beta_1$ decreases gradually when the duration of sunlight is higher than 10 hours. This result is consistent with previous studies. Excessive sunshine hours can lead to high temperatures and lower human travel rates, and population density is one of the key factors in the spread of SFTS [22].

Therefore, we believe that daylight is one of the important factors affecting SFTS transmission, mainly affecting animal-to-human and human-to-human transmission routes, and that the activity time of animals kept outside should be limited as much as possible [23].

Compared with the duration of sunshine, relative humidity was also one of the factors that affect the incidence of SFTS reports. According to previous studies, the wet days may affect the metabolic rate and reproductive capacity of ticks [9]. The results of this study indicate that relative humidity had a small effect on the ability of ticks to transmit to people. This may be because we only studied the process of tick's transmission, rather than the relationship between relative humidity and ticks themselves. In addition, the relative humidity has a more obvious influence on $\beta_{31}$, and it increases with the increase of relative humidity. It may be due to the increase in relative humidity that enhances the reproductive capacity and metabolic rate of ticks that live on the host animal, plus the long sunshine duration [24]. The increase leads to an increase in the surface temperature of the host animals. As shown by previous studies, the temperature is also one of the factors affecting the growth and development of ticks [25].

As we know, temperature plays an important role in vector-borne diseases, which can affect the replication and transmission of pathogens. The faster the virus multiplies, the fewer times it needs to be transmitted to animals and humans [26]. Many studies have used the distributed time lag nonlinear model to explore the time lag effect of temperature, indicating that the incidence of SFTS is highest when the temperature is 20–24˚C. This is the same as the results of this study. The results of this study also indicate that a temperature ($>$26˚C) will inhibit the reported incidence of SFTS. This may be affected by two main reasons. First, the transmissibility between people will increase when the temperature increases, which means the temperature suitable for the reproduction and growth of ticks was also just suitable for humans to travel and work. Therefore, when the temperature is lower than 30˚C, human-to-human contact will cause the spread of SFTS more frequently. Secondly, the host animal's ability to transmit to humans also remains stable at 20–25˚C. When the temperature is too high, $\beta_{31}$ will also decreased, which can also prove the impact of temperature on the reproduction of ticks and human travel [27]. Therefore, in fact, temperature really affects humans and host animals in the transmission dynamics of SFTS. The SFTS epidemic warning is issued at 20–25˚C to remind people that they should take protective measures as much as possible when going out to work, or reduce travel and tourism. For the host animals, it is essential to eliminate ticks on the body surface to reduce the chance of infection due to direct contact with ticks [9].

In this study, wind speed and precipitation had little effect on the incidence rate and the infection rate coefficient of each transmission route, although previous studies have speculated that wind speed will increase the diffusion of carbon dioxide and increase the mortality of ticks, thereby affecting the occurrence of SFTS [28]. However, the research evidence on wind speed and precipitation was rare, and the only research was the same as our research results, and it is relatively insensitive to the occurrence of SFTS and rainfall [12].

In addition to meteorological factors, this study introduced month and tick density into GAM, and found that both had a certain contribution to different GAMs. As far as we know, SFTS is a seasonal disease, which usually occurs in summer and autumn [29]. But for the infection rate coefficients of different transmission routes, the peak of infection of $\beta_1$, $\beta_{21}$ and $\beta_{31}$ was generally from May to September, while the transmission of the environment was mainly in winter. This may be because the virus transmission in the environment requires a certain time lag. Based on the peak of infection of various transmission routes, the peak of SFTS incidence shown in our report was mainly from May to October reported by existing researches [30]. In this study, the influence of tick density on each transmission route and incidence rate was non-linear, but it is worth noting that the influence of tick density on $\beta_{21}$ was small, which may be due to the consideration of the route of tick transmission in the modeling process. It is

not comprehensive enough, but from the reported incidence rate and other infection rate coefficients, the tick density will also have two peaks, which were 20*flags/hour and 45* flags/hour. We should strengthen tick monitoring, set warning thresholds, remind people to pay attention to areas with high tick density, and even set up warning areas for perennial monitoring to reduce the occurrence of SFTS.

By establishing the GAM of meteorological factors and tick density on the incidence of SFTS and the transmission rate coefficients of different routes, combined with meteorological characteristics of Jiangsu Province, we also explored the changes in the occurrence of SFTS under extreme weather conditions. In this study, we found that extreme conditions of precipitation and wind speed caused a sharp decrease in the incidence of SFTS in the short term, which may be due to the restriction of human activities due to hurricanes and the simultaneous decrease in tick density. Although the GAM showed that temperature and sunshine duration were important factors influencing the occurrence of SFTS, the fact that both temperature and sunshine duration were at extreme values simulating drought conditions had less effect on the occurrence of SFTS, probably due to the large values taken for temperature and sunshine duration.

After we performed simulations of the interventions, it was found that the most important thing was to control the transmission from the environment to humans, which is consistent with the results of previous studies. This may be due to the inability to completely separate each transmission route when modeling transmission dynamics. Transmission in the environment is also a newly proposed transmission route, and it is difficult to abstract out the environmental ticks to consider the effect of the environment separately.

In summary, although the simulation results are poor for the interventions according to MMDM, this may be due to the difficulty of simulating pure transmission dynamics models without real prevention and control data. However, we can also find that controlling human-environmental exposure can lead to a reduction in the incidence of SFTS. Therefore, combining the transmission kinetic model with the distribution of SFTS incidence areas and the relationship between various transmission routes and meteorological factors, we can still make the following recommendations for SFTS prevention and control: 1. SFTS has a high incidence in hilly or forested areas, so we should focus on the people living in these areas and do a good job in health education. 2. Avoid working outside for too long when it's hot, and also to do a good job of temperature threshold warning and avoid going outside as much as possible.3. To control ticks, it is important not only to monitor the density of ticks in each area, but also to eliminate ticks on the surface of livestock and animals to reduce the chance of direct contact and transmission.4. When working in a field environment, one should do a good job of self-protection and minimize direct contact.

## Limitations

This study was based on the MMDM established in the previous stage to calculate the infection rate coefficient of each transmission route. Because the multi-population and multi-route dynamics model was more complicated, the factors considered may not be perfect, so the basic reproduction number of SFTS was not calculated. The generalized additive model used in this study requires certain professional background knowledge for the interpretation of the results, and certain subjective judgments may appear. The validation for Zhejiang Province may be biased because no data on tick density are available for Zhejiang Province, which is the result of our separate simulation of a generalized additive model of tick density and meteorological factors. When performing simulations of interventions, there may be some errors due to the complexity of SFTS transmission.

## Conclusions

As is well known, this study is the first to date to explore SFTS transmission by exploring the relationship among different transmissibility, meteorological factors and tick density. Our results show that sunshine duration, temperature and relative humidity are key meteorological factors for SFTS transmission, and different meteorological factors affect different transmission rate (multi-population multi-route transmission). Hurricanes reduce the incidence of SFTS in the short term, but have little effect in the long term. The most useful interventions against SFTS are to reduce the exposure of the population to high-risk environments, while tick density is an important factor influencing the occurrence of SFTS, and monitoring systems should be further improved.

## Supporting information

**S1 Fig. SFTS reported incidence rate and coefficient of infection rate by different transmission routes over time in Jiangsu Province, 2017–2019.** A: Reported incidence rate changes over time; B: Coefficient of environmental-to-human transmission of infection rate changes over time; $\beta_{w1}$: Coefficient of environmental-to-human transmission of infection rate; C: Coefficient of human-to-human transmission of infection rate changes over time; $\beta_{11}$: Coefficient of human-to-human transmission of infection rate; D: Coefficient of animal-to-human transmission of infection rate changes over time; $\beta_{31}$: Coefficient of animal-to-human transmission of infection rate; E: Coefficient of tick-to-human transmission of infection rate changes over time; $\beta_{21}$: Coefficient of tick-to-human transmission of infection rate.
(TIF)

**S2 Fig. Meteorological factors and tick density changes over time in Jiangsu Province, 2017–2019.** A: Relative humidity over time in Jiangsu Province; B: Precipitation changes over time; C: Sunshine duration changes over time; D: Average temperature changes over time; E: Tick density changes over time; F: Wind speed changes over time.
(TIF)

**S3 Fig. Non-linear relationship between SFTS incidence with meteorological factors and tick density in Jiangsu Province.** Part A: SFTS incidence with meteorological factors and tick density in no time lag GAM; Part B: SFTS incidence with meteorological factors and tick density in time lag GAM; SD = Sunshine duration; AT = Average temperature; WS = Wind speed; TD = Tick density.
(TIF)

**S4 Fig. Non-linear relationship between human-to-human transmissibility with meteorological factors and tick density in Jiangsu Province.** Part A: human-to-human transmissibility with meteorological factors and tick density in no time lag GAM; Part B: human-to-human transmissibility with meteorological factors and tick density in time lag GAM; SD = Sunshine duration; RH = Relative humidity; AT = Average temperature; PRCP = 24-hour precipitation; WS = Wind speed; TD = Tick density.
(TIF)

**S5 Fig. Non-linear relationship between environment-to-human transmissibility with meteorological factors and tick density in Jiangsu Province.** Part A: environment-to-human transmissibility with meteorological factors and tick density in no time lag GAM; Part B: environment-to-human transmissibility with meteorological factors and tick density in time lag GAM; SD = Sunshine duration; RH = Relative humidity; AT = Average temperature; PRCP = 24-hour precipitation; WS = Wind speed; TD = Tick density.
(TIF)

**S6 Fig. Non-linear relationship between tick-to-human transmissibility with meteorological factors and tick density in Jiangsu Province.** Part A: tick -to-human transmissibility with meteorological factors and tick density in no time lag GAM; Part B: tick-to-human transmissibility with meteorological factors and tick density in time lag GAM; SD = Sunshine duration; RH = Relative humidity; AT = Average temperature; PRCP = 24-hour precipitation; WS = Wind speed; TD = Tick density.
(TIF)

**S7 Fig. Non-linear relationship between animal-to-human transmissibility with meteorological factors and tick density in Jiangsu Province.** Part A: animal-to-human transmissibility with meteorological factors and tick density in no time lag GAM; Part B: animal-to-human transmissibility with meteorological factors and tick density in time lag GAM; SD = Sunshine duration; RH = Relative humidity; AT = Average temperature; PRCP = 24-hour precipitation; WS = Wind speed; TD = Tick density.
(TIF)

**S1 Table. The analyzed data in this study.**
(XLSX)

## Acknowledgments

We thank the staff members at the hospitals, local health departments, and municipal and county-level Center for Disease Control and Prevention offices for their valuable assistance in coordinating the data collection. We also thank Dr. Xiao Wupeng, college of Environment and Ecology, Xiamen University. Thanks to Qingqing Hu, a PHD at the University of Utah, for helping with the English language touch-ups on this article.

## Author Contributions

**Conceptualization:** Bin Deng, Jia Rui, Shengnan Lin, Jingwen Xu, Jiefeng Huang, Yao Wang, Jian-li Hu, Tianmu Chen.

**Data curation:** Bin Deng, Shu-yi Liang, Zhi-feng Li, Kangguo Li, Weikang Liu, Hongjie Wei, Chan Liu, Zhuoyang Li, Peihua Li, Meng Yang, Yuanzhao Zhu, Xingchun Liu, Nan Zhang, Xiao-qing Cheng, Jian-li Hu.

**Formal analysis:** Bin Deng, Shu-yi Liang, Kangguo Li, Shengnan Lin, Li Luo, Hongjie Wei, Tianlong Yang, Yao Wang, Meng Yang, Yuanzhao Zhu, Xingchun Liu, Xiao-chen Wang.

**Funding acquisition:** Tianmu Chen.

**Investigation:** Bin Deng, Jia Rui, Shu-yi Liang, Zhi-feng Li, Jiefeng Huang, Tianlong Yang, Zhuoyang Li, Peihua Li, Nan Zhang, Xiao-qing Cheng, Xiao-chen Wang, Jian-li Hu, Tianmu Chen.

**Methodology:** Bin Deng, Kangguo Li, Shengnan Lin, Jingwen Xu, Tianlong Yang, Jian-li Hu, Tianmu Chen.

**Project administration:** Shu-yi Liang, Tianmu Chen.

**Resources:** Jia Rui, Shu-yi Liang, Zeyu Zhao, Nan Zhang, Jian-li Hu, Tianmu Chen.

**Software:** Bin Deng, Shengnan Lin, Li Luo, Weikang Liu, Chan Liu, Zeyu Zhao, Jian-li Hu.

**Supervision:** Jian-li Hu, Tianmu Chen.

**Validation:** Shu-yi Liang, Li Luo, Weikang Liu, Jiefeng Huang, Hongjie Wei, Chan Liu, Zhuoyang Li, Peihua Li, Yao Wang, Meng Yang, Tianmu Chen.

**Visualization:** Bin Deng, Li Luo.

**Writing – original draft:** Bin Deng, Shu-yi Liang.

**Writing – review & editing:** Bin Deng, Jia Rui, Jian-li Hu, Tianmu Chen.

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
