## [Decision Letter · Decision Letter 0]

4 Jan 2022

Dear Dr. Chen,

Thank you very much for submitting your manuscript "Meteorological factors and tick density affect the dynamics of SFTS in Jiangsu Province, China" for consideration at PLOS Neglected Tropical Diseases. As with all papers reviewed by the journal, your manuscript was reviewed by members of the editorial board and by several independent reviewers. In light of the reviews (below this email), we would like to invite the resubmission of a significantly-revised version that takes into account the reviewers' comments. 

- The language needs to be checked carefully.

- Please upload the analyzed data as a supplementary file for easy reproducibility of the reported findings.

- Please write the section name and lines/pages number for every change you make.

We cannot make any decision about publication until we have seen the revised manuscript and your response to the reviewers' comments. Your revised manuscript is also likely to be sent to reviewers for further evaluation.

Sincerely,

Mohamed Gomaa Kamel

Associate Editor

Nam-Hyuk Cho

Deputy Editor

- The language needs to be checked carefully.

- Please upload the analyzed data as a supplementary file for easy reproducibility of the reported findings.

- Please write the section name and lines/pages number for every change you make.

Reviewer's Responses to Questions

**Key Review Criteria Required for Acceptance?**

**Methods**

-Are the objectives of the study clearly articulated with a clear testable hypothesis stated?

-Is the study design appropriate to address the stated objectives?

-Is the population clearly described and appropriate for the hypothesis being tested?

-Is the sample size sufficient to ensure adequate power to address the hypothesis being tested?

-Were correct statistical analysis used to support conclusions?

-Are there concerns about ethical or regulatory requirements being met?

Reviewer #1: Bin Deng et al used multi-population and multi-route dynamic model aimed to explore whether the transmission routes of severe fever with thrombocytopenia syndrome (SFTS) will be affectedby tick density and meteorological factors, and to explore the factors that affect the transmission of SFTS，However, the data and analysis in the article are not clear about the conclusions obtained.

Reviewer #2: -Are the objectives of the study clearly articulated with a clear testable hypothesis stated?

Yes

-Is the study design appropriate to address the stated objectives?

Yes

-Is the population clearly described and appropriate for the hypothesis being tested?

Yes

-Is the sample size sufficient to ensure adequate power to address the hypothesis being tested?

Yes

-Were correct statistical analysis used to support conclusions?

Yes

-Are there concerns about ethical or regulatory requirements being met?

No ethical or regulatory concerns

Reviewer #3: (No Response)

**Results**

-Does the analysis presented match the analysis plan?

-Are the results clearly and completely presented?

-Are the figures (Tables, Images) of sufficient quality for clarity?

Reviewer #1: the objective of this study is to obtain meteorological factors and tick density affect the infected dynamics of SFTS factors through modeling, not the affect about transmission routes.

Reviewer #2: -Does the analysis presented match the analysis plan?

Yes

-Are the results clearly and completely presented?

No. The authors need to establish the generalizability of the model and make meaningful predictions using the model. The following major concerns need to be addressed:

1. GAMs are known to have the limitation of over-fitting. The authors need to calculate incidence and transmission of SFTS using an independent dataset, possibly from another province. If the model is found to be over-fitting, the authors need to more carefully select the independent meteorological variables since many of them were found to be significantly correlated with each other. They might also employ one of the dimensionality reduction techniques.

2. The generalizability of the model need to be established by making predictions about national incidence and transmission rates employing the model.

3. Although the authors have made recommendations about public health interventions for control and prevention of SFTS, they need to make predictions about how these interventions could impact SFTS incidence and transmission. Also they can make predictions about the possible impact of global climate change as well as regional natural calamities such as hurricanes, heavy rains or floods can have of SFTS incidence and transmission.

-Are the figures (Tables, Images) of sufficient quality for clarity?

The figures need to be more self-explanatory. The labels are not legible in Figures 4, 5, 8 and 9.

Reviewer #3: (No Response)

**Conclusions**

-Are the conclusions supported by the data presented?

-Are the limitations of analysis clearly described?

-Do the authors discuss how these data can be helpful to advance our understanding of the topic under study?

-Is public health relevance addressed?

Reviewer #1: some issues should be explained

1,Please explain the definitions of the four types of transmission: human-to-human transmission, tick-to-human transmission, environmental transmission and host animal transmission. 

2,Usually, human-to-human transmission of SFTS is rare, and the transmission routes of the infected cases is not reported in this study.

Reviewer #2: -Are the conclusions supported by the data presented?

Yes

-Are the limitations of analysis clearly described?

Yes

-Do the authors discuss how these data can be helpful to advance our understanding of the topic under study?

No. The authors have not made significant efforts is making predictions about the changes in natural history of disease with global climate change and regional climate disasters

-Is public health relevance addressed?

Yes. However, they do not employ their model to predict the impact of public health interventions could have on SFTS incidence and transmission, and recommend the most-effective interventions that can be implemented at the regional or national level.

Reviewer #3: (No Response)

**Editorial and Data Presentation Modifications?**

Reviewer #1: Major revision

Reviewer #2: The figures need to be more self-explanatory. The labels are not legible in Figures 4, 5, 8 and 9. The quality and presentation of the figures need significant improvement. The manuscript has to be proof-read by an English language expert.

Reviewer #3: Correct term "a new Bunya virus" and " a new Bunia virus infection" (str. 71, p. 4; str 85, p 5). Severe fever with thrombocytopenia syndrome (SFTS) is an emerging infectious disease caused by Dabie bandavirus also known as the SFTS virus (Riboviria-Orthornavirae-Negarnaviricota-Polyploviricotina-Ellioviricetes-Bunyavirales-Phenuiviridae-Bandavirus-Dabie bandavirus)

**Summary and General Comments**

Reviewer #1: some issues should be explained

1,Please explain the definitions of the four types of transmission: human-to-human transmission, tick-to-human transmission, environmental transmission and host animal transmission. 

2,Usually, human-to-human transmission of SFTS is rare, and the transmission routes of the infected cases is not reported in this study. 

3, the objective of this study is to obtain meteorological factors and tick density affect the infected dynamics of SFTS factors through modeling, not the affect about transmission routes. 

4,Figure 4-9 is unclear should be load again

Reviewer #2: The authors use a multi-population and multi-route dynamic model established previously to calculate the infection rate coefficients of various transmission routes of SFTS in Jiangsu Province of China, and a generalized additive model was established to further elaborate the influence of meteorological factors and tick density on SFTS incidence and transmission. The study is well-conducted, analyzed and interpreted. However, the readership and impact of the study will be greatly benefited if the following concerns are addressed.

Major comments

1. GAMs are known to have the limitation of over-fitting. The authors need to calculate incidence and transmission of SFTS using an independent dataset, possibly from another province. If the model is found to be over-fitting, the authors need to more carefully select the independent meteorological variables since many of them were found to be significantly correlated with each other. They might also employ one of the dimensionality reduction techniques.

2. The generalizability of the model need to be established by making predictions about national incidence and transmission rates of SFTS employing the model.

3. Although the authors have made recommendations about public health interventions for control and prevention of SFTS, they need to make predictions about how these interventions could impact SFTS incidence and transmission. Also they can make predictions about the possible impact of global climate change as well as regional natural calamities such as hurricanes, heavy rains or floods can have of SFTS incidence and transmission.

Minor Comments

1. The figures need to be more self-explanatory. 

2. The labels are not legible in Figures 4, 5, 8 and 9. 

3. The quality and presentation of the figures need significant improvement. 

4. The manuscript has to be proof-read by an English language expert.

Reviewer #3: The paper by Deng et al. describes the influence of different meteorological factors (wind speed, duration of sunshine, average temperature, amount of precipitation, atmospheric pressure and relative humidity) and ticks density on the incidence of f severe fever with thrombocytopenia syndrome (SFTS).

The relationship between all meteorological factors, tick density and was described by statistical method (Spearmen test). The relationship between SFTS incidence rates, SFTS transmission rate and different meteorological indicators and ticks density was assessed by constructing generalized additive models (GAMs) in in R 3.2.3. The choice of the most suitable model was determined based on the generalized cross-validation score (GCV).

It has been shown that the greatest influence on the incidence rate is exerted by the sunshine duration, temperature and relative humidity.

The paper is well illustrated, the conclusions are supported by the obtained results.

The new data obtained in the work can be used to predict the SFTS incidence.

Changes should be made to improve the manuscript:

1. Describe the epidemiology of SFTS in China, incl. describe the main route of human infection in Jiangsu Province from 2017 to 2019. When constructing models for various mechanisms of infection (β1, β21, β31 and βw1), were all cases of human disease taken into account or only with a certain route of infection? Explain in more detail the epidemiological meaning of transmission routes β1, β21, β31 and βw1.

2.Describe what species of ticks are vectors of SFTS virus in China. Was the ticks species composition taken into account when assessing the number of ticks and their effect on the incidence rate?

PLOS authors have the option to publish the peer review history of their article (what does this mean?). If published, this will include your full peer review and any attached files.

Reviewer #1: No

Reviewer #2: No

Reviewer #3: No
---

## [Decision Letter · Decision Letter 1]

19 Apr 2022

Dear Dr. Chen,

We are pleased to inform you that your manuscript 'Meteorological factors and tick density affect the dynamics of SFTS in Jiangsu Province, China' has been provisionally accepted for publication in PLOS Neglected Tropical Diseases.

Best regards,

Mohamed Gomaa Kamel

Associate Editor

Nam-Hyuk Cho

Deputy Editor

Reviewer's Responses to Questions

**Key Review Criteria Required for Acceptance?**

**Methods**

-Are the objectives of the study clearly articulated with a clear testable hypothesis stated?

-Is the study design appropriate to address the stated objectives?

-Is the population clearly described and appropriate for the hypothesis being tested?

-Is the sample size sufficient to ensure adequate power to address the hypothesis being tested?

-Were correct statistical analysis used to support conclusions?

-Are there concerns about ethical or regulatory requirements being met?

Reviewer #1: (No Response)

Reviewer #2: No further comments

Reviewer #3: -Are the objectives of the study clearly articulated with a clear testable hypothesis stated?

Yes

-Is the study design appropriate to address the stated objectives?

Yes

-Is the sample size sufficient to ensure adequate power to address the hypothesis being tested?

Yes

-Were correct statistical analysis used to support conclusions?

Yes

-Are there concerns about ethical or regulatory requirements being met?

No

**Results**

-Does the analysis presented match the analysis plan?

-Are the results clearly and completely presented?

-Are the figures (Tables, Images) of sufficient quality for clarity?

Reviewer #1: (No Response)

Reviewer #2: No further comments

Reviewer #3: -Does the analysis presented match the analysis plan?

Yes

-Are the results clearly and completely presented?

Yes

-Are the figures (Tables, Images) of sufficient quality for clarity?

Yes

**Conclusions**

-Are the conclusions supported by the data presented?

-Are the limitations of analysis clearly described?

-Do the authors discuss how these data can be helpful to advance our understanding of the topic under study?

-Is public health relevance addressed?

Reviewer #1: (No Response)

Reviewer #2: No further comments

Reviewer #3: -Are the conclusions supported by the data presented?

Yes

-Are the limitations of analysis clearly described?

Yes

-Do the authors discuss how these data can be helpful to advance our understanding of the topic under study?

Yes

-Is public health relevance addressed?

Yes

**Editorial and Data Presentation Modifications?**

Reviewer #1: (No Response)

Reviewer #2: The authors have addressed all the concerns and the manuscript can be accepted for publication.

Reviewer #3: Accept

**Summary and General Comments**

Reviewer #1: (No Response)

Reviewer #2: The authors have addressed all the concerns and the manuscript can be accepted for publication.

Reviewer #3: The manuscript has improved a lot after revision. The authors have addressed all of my comments. I now would like to support the publication of the current form of the manuscript

PLOS authors have the option to publish the peer review history of their article (what does this mean?). If published, this will include your full peer review and any attached files.

Reviewer #1: No

Reviewer #2: No

Reviewer #3: No

---

## [Editor Report · Acceptance letter]

5 May 2022

Dear Mr. Chen,

We are delighted to inform you that your manuscript, "Meteorological factors and tick density affect the dynamics of SFTS in Jiangsu Province, China," has been formally accepted for publication in PLOS Neglected Tropical Diseases.

Best regards,

Shaden Kamhawi

co-Editor-in-Chief

Paul Brindley

co-Editor-in-Chief
